# The organisation of primary health care service delivery for non-communicable diseases in Nigeria: A case-study analysis

**Whenayon Simeon Ajisegiri**[1]*, **Seye Abimbola**[1,2], **Azeb Gebresilassie Tesema**[1,3], **Olumuyiwa O. Odusanya**[4], **David Peiris**[1‡], **Rohina Joshi**[5,6‡]

1 The George Institute for Global Health, University of New South Wales (UNSW), Sydney, Australia, 2 School of Public Health, University of Sydney, Sydney, Australia, 3 School of Public Health, Mekelle University, Mekelle, Ethiopia, 4 Department of Community Health and Primary Health Care, Lagos State University College of Medicine, Ikeja, Nigeria, 5 School of Population Health, University of New South Wales (UNSW), Sydney, Australia, 6 The George Institute for Global Health, New Delhi, India

‡ DP and RJ equal senior authors on this work.
* wajisegiri@georgeinstitute.org.au

**Data Availability Statement:** All relevant data contributing to the findings are within the study. All raw quantitative data from the surveys, and

## Abstract

As chronic diseases, non-communicable diseases (NCDs) require sustained person-centred and community-based care. Given its direct link to communities and households, Primary Health Care (PHC) is well positioned to achieve such care. In Nigeria, the national government has prioritized PHC system strengthening as a means of achieving national NCD targets. However, strengthening PHC systems for NCDs require re-organization of PHC service delivery, based on contextual understanding of existing facilitators and barriers to PHC service delivery for NCDs. We conducted a mixed method case study to explore NCD service delivery with 13 PHC facilities serving as the cases of interest. The study was conducted in two northern and two southern states in Nigeria–and included qualitative interviews with 25 participants, 13 focus group discussion among 107 participants and direct observation at the 13 PHCs. We found that interprofessional role conflict among healthcare workers, perverse incentives to sustain the functioning of PHC facilities in the face of government under-investment, and the perception of PHC as an inferior health system were major barriers to improved organisation of NCD management. Conversely, the presence of physicians at PHC facilities and involvement of civil society organizations in aiding community linkage were key enablers. These marked differences in performance and capacity between PHC facilities in northern compared to southern states, with those in the south better organised to deliver NCD services. PHC reforms that are tailored to the socio-political and economic variations across Nigeria are needed to improve capacity to address NCDs.

## Introduction

In Nigeria, NCDs account for 29% of all deaths, out of which cardiovascular diseases accounts for 11% [1]. The prevalence of hypertension and diabetes is estimated to be 28.9% and 4.1% respectively [2]. In line with the Global Action Plans [3] and Sustainable Development Goal 3.4 [4] (to strengthen responses for prevention and control of NCDs), Nigeria has also set

summary of findings from qualitative data have been anonymised and uploaded as supplementary files (MS Excel and MS Word documents respectively).

**Funding:** The project was supported by the George Institute for Global Health, Australia through the Seed Grant funds dedicated for under-served populations in LMICs for 2019/2020. The UNSW Scientia Scholarship program supports WSA and AGT. WSA is also supported through the Australian Government Research Training Program Scholarship. SA was supported by the Australian National Health and Medical Research Council (NHMRC) through an Overseas Early Career Fellowship (APP1139631). RJ is supported by the Australian National Heart Foundation (APP 102059) and UNSW Scientia Fellowship. DP is supported by NHMRC career Development Fellowship, Level 2 and Australia National Heart Foundation Future Leader Fellowship. The funders had no role in study design, data collection and analysis, decision to publish, or preparation of the manuscript.

**Competing interests:** The authors declare no competing interest.

national NCD targets. These include relative reduction of raised blood pressure and diabetes mellitus by 25% in year 2025 [1]. However, as recommended in World Health Organization's (WHO) "best-buys", effective and feasible implementation of NCD prevention and control strategies in low- and middle-income countries (LMICs), require strengthening and orientating the health system through people-centred primary health care (PHC) [5].

Nigeria's PHC system, the bedrock of the health system [6], was adopted into the 1988 national health policy in an attempt to improve access and utilization of basic health services [7]. Between 1986 to 1990, the PHC experienced several reforms including its expansion to all LGAs and the creation of colleges of health technology for the training of community health workers [8]. In 2014, the National Health Act placed the PHCs under the local government authorities [9]. This however created a bottleneck for PHC funding as federal laws are not binding on the states in a decentralized health system [8].

After several evolutions, the current strategic drive for the PHC is the ward health system (WHS). It was developed by the National Primary Health Care Development Agency (NPHCDA) in order to improve access to healthcare [10]. The WHS comprises of several interventions known as the Ward Health Minimum Package. These interventions are expected to address communicable, non-communicable diseases, and maternal and child health services. [11]. The direct linkage of the PHC to communities and households positions it as the pedestal to reach the 'last mile' population who are mostly located in rural areas [12]. Despite its pivotal role in addressing access barriers, PHC attracts the least investment in the national health system [6]. This is in part due to the devolution of PHC to the local government, the level of governance with the weakest technical and financial capacity.

Due to the chronic nature of NCDs, people living with these diseases and their risk factors require sustained person-centred and community-based care [13]. This would ideally be achieved through the PHC health system. However, NCD care appears to be the most neglected aspect of the PHC sector as comprehensive NCD care is omitted from the Minimum Package of Health Services that will be funded by the Basic Health Care Provision Fund [1]. This may partly be due to the fact that the WHS was established to align with Millennium Development Goals which largely omitted NCDs as it was perceived to contribute towards a proportionately smaller burden of disease at that time [7]. It may also be because most government and development partners' interventions to strengthen PHC are focused mainly on maternal, child and reproductive health as well as infectious diseases [11], still aligning with the primary reason for establishing PHC systems especially in the 1980s [6].

Access to quality and essential NCD interventions is further compounded by inadequate and maldistribution of skilled health workers, particularly physicians and nurses [13]. Consequently, NCD care at the PHC level is mainly handled by community health workers (CHWs), whose training and skills are generally considered insufficient for NCD management and prevention [14]. Indeed, WHO's country profile on NCDs reported that no PHC facility in Nigeria offers cardiovascular disease (CVD) risk stratification or utilizes CVD guidelines; only 30% of facilities reported essential NCD medicines as "generally available"; and similarly only 33% reported essential NCD technologies as "generally available" [15]. Given the national government's commitment to prioritize PHC system strengthening as a means of achieving universal health coverage [6] and national NCD targets [1], a corresponding organizational restructuring of service delivery is critical. Such re-organisation requires contextual understanding of existing organisation and the facilitators and barriers to PHC service delivery for NCDs across different settings in Nigeria.

Most studies on NCDs in Nigeria centre on disease burden [2, 16–19] while some others have assessed health care workers' knowledge on these diseases [13, 20]. Only few studies have examined enablers and barriers to NCD service delivery in PHC facilities. One study revealed

that NCDs have the lowest service-readiness scores with major gaps in staff capacity to treat NCDs [8]. Another study among PHC health workers and insurance managers, showed that health insurance was perceived as an important facilitator of implementing high-quality hypertension care; while high staff workload; administrative challenges; and difficulty in adapting some guideline recommendations were key inhibitors of high-quality care [21]. In another study, availability of drugs at subsidized rates, trained workforce and regular training opportunities were identified as factors promoting quality; while cultural barriers and patients' socioeconomic factors were identified as major barriers to receiving high quality care [22].

While identifying enablers and barriers for quality PHC service delivery for NCDs is important, national and sub-national decision-making requires a nuanced understanding of the context and structure within which these services are delivered. In this study, we sought to characterise the organisation of PHC service delivery for NCDs and identify what factors promote or hinder NCD service delivery, with special focus on hypertension and diabetes mellitus. We define such factors (i.e., enablers or barriers) as the people, processes, structures, skills and strategies that may facilitate or constrain high-quality NCD service delivery at the PHC level.

## Materials and methods

We conducted a mixed-method case study to explore NCD service delivery with each PHC facility serving as the case.

### Study setting

Three PHC facility types exist based on the Ward Health System [23]:

1. **Health Posts (Dispensary):** It has a population coverage of 500 and operates for 8 hours daily with one junior community health extension worker (JCHEW). The JCHEW is also expected to spend 60% of his/her time in the community where they work with and supervise community resource persons like the Traditional Birth Attendants. It is aimed at treating minor ailment and encouraging pregnant women to register for ante-natal services.

2. **Primary Health Clinic (Maternity Centre, Basic Health Centre):** This has a population coverage of 2000–5000 and operates on a 24-hour service. It is expected to have a midwife who handles maternal health including delivery, 4 JCHEW and 2 CHEW who provide health services both within the PHC and the community.

3. **The Primary Health Care Centre (Comprehensive Health Centre, Model PHC Centre):** This is the highest-level facility type with a population coverage of 10,000–20,000 people, and the political ward is its service delivery area. The PHC centre operates on a 24-hour basis and is expected to be staffed with all cadres of CHWs, nurses/midwife, doctors (if available), pharmacy technicians, an environmental health officer, medical records officer, laboratory technician and other support staff. It equipped and staffed to for higher diagnostic capacity, provide basic emergency obstetric cares and treat more ailments than the afore-mentioned centres.

Our study was conducted mainly at the highest level of the PHC system, the Primary Health Care Centres, where the most comprehensive form of service delivery with well-equipped and adequately skilled staff is expected.

### Study design

This case study used mixed methods (survey and interviews) to identify enablers and barriers to NCD (hypertension and diabetes mellitus) diagnosis and management at PHC level. A case

study is a pragmatic design that seeks to explore contemporary phenomena in real-world settings [24]. It has potential to provide in-depth answers to the "how" and "why" of NCD service delivery at the facility level.

### Study area

A total of 13 PHC centres in four Nigerian states participated in the study. Two states were purposively selected in each of the northern and southern regions to reflect the regional socio-political, economic, and religious differences that are known to influence healthcare demands, household health seeking behaviours, and availability of medication and equipment in health facilities [25, 26]. An additional criterion for selecting two states (one in northern and one in southern Nigeria) was based on recent implementation of a new health intervention programme (performance-based financing) in those states, designed to re-organise PHC service delivery.

### Quantitative data collection

The WHO Service Availability and Readiness Assessment (SARA) tool was used to collect data on services available for NCDs (hypertension and diabetes) at each PHC facility [27]. The tool was adapted for the hypertension programme in Nigeria, with input from government agencies and relevant stakeholders. The adaptation specifically focused on diagnosis and management of hypertension and diabetes mellitus. It was subsequently pilot tested and data collected in 60 PHC facilities [28]. Data collection focussed on: (i.) service availability (ii) patient access, (iii) staffing capacity, (iv) infrastructure, (v) basic client amenities, (vi) infection control, (vii) healthcare waste management, (viii) clinical mentoring, (ix) basic equipment, (x) available services for non-communicable diseases and diagnostics, (xi) supply chain, (xii) medicines and vaccines, and (xiii) commodities [28]. This corresponded to the 13 sections of the SARA tool. In addition, data collected with the SARA survey instrument from PHC respondents were corroborated by direct observation by research staff of medications, equipment, and supplies in the PHC facilities.

### Qualitative data collection

In each PHC, in-depth, semi-structured interviews (IDI) with key PHC staff (nurses, community health workers or doctors) and focus group discussions (FGD) with about 6–10 participants were conducted among health workers who have worked for a minimum of three months at the facility. This is to ensure participants had worked for a sufficient duration to have detailed understanding of how their facilities operate. The focus of the interviews was to understand the factors that constitute barriers and enablers (defined below) to NCD management. All participants were interviewed face-to-face at their respective PHC facilities. Interview durations ranged from 35–60 minutes while FGDs last for 45–70 minutes. Both interviews and FGDs were audio-recorded, transcribed and field notes were also taken.

Data were collected from August 2019 to September 2019 guided by the consolidated criteria for reporting qualitative research guidelines for qualitative research [29].

### Data analysis

The unit of analysis in the study was a case as represented by each PHC facility. The quantitative data for each case was analyzed first to provide a general overview of each case. Findings from the quantitative data guided the initial analysis of the qualitative data.

**Quantitative data.** Statistical analysis was done in Microsoft Excel and continuous, non-parametric measures were summarized by median and interquartile range. The facility-based service availability for hypertension and diabetes and other domains of interest such as equipment and supplies, personnel and medications were tabulated as frequencies.

**Qualitative data.** Verbatim transcription of recorded FGDs and interviews was done. These transcripts were imported into NVivo 12 for data coding, which used the pre-determined coding from SARA results. The codes were categorized based on how related they were. Several meetings were subsequently held by the research team members to analyse and interpret the data from each case. These meetings helped to iteratively identify and refine emerging themes, and inferences, and to deal with apparently contradictory information across the cases. Phrases or full sentences that most accurately expressed or illustrated the categories under each theme were then identified and presented as quotes in the results section.

## Ethical considerations

Ethical approval was granted by the National Health Research Ethics Committee of Nigeria (Approval no: NHREC/01/01/2007) and the University of New South Wales Human Research Ethics Committee (HC: 190051). Informed written consent was obtained from all participants before conducting the interview. Anonymity and confidentiality of all respondents were maintained throughout the process. Participants names and the names of the states and PHC facilities included in the study were replaced with codes during data analysis (Table 1).

## Results

The case study analysis described available services and explored the potential barriers and enablers to service delivery. A more descriptive information on each of the cases is provided in S1 Annex. Based on findings from our study, Fig 1 and Table 2 respectively provides information on the combined and disaggregated SARA findings of NCD-related services at the PHC facilities.

### PHC team composition and capacity

Physicians are present in five facilities, while 3 facilities had neither physician nor nurse. Five facilities had staff who had been trained in screening or management of NCDs within the

**Table 1. Participants of key informant interview (KII) and focused group discussion FGD).**

|  | Region | KII1 | KII2 | KII3 | KII 4 | FGD participants |
|---|---|---|---|---|---|---|
| Case 1 | North | M, 54, CHEW | F, 42, Nurse | – | – | 4 M, 6 F |
| Case 2 | North | F, 40, CHO | F, 52, Nurse | – | – | 5 F |
| Case 3 | North | F, 45/, CHO | F, 52, Nurse | – | – | 3 M, 6 F |
| Case 4 | North | F, 45, CHEW | – | – | – | 2 M, 6 F |
| Case 5 | North | F, 49, CHO | – | – | – | 1 M, 5 F |
| Case 6 | North | F, 55, CHO | – | – | – | 4 M, 6 F |
| Case 7 | North | F, 48, CHO | – | – | – | 2 M, 4 F |
| Case 8 | South | M, 55, CHO | F, 34, Nurse | F, 40, Physician | M, 47, Medical Officer of Health | 4 F, 6 F |
| Case 9 | South | M, 55, CHO | F, 47, Nurse | F, 37, Physician | – | 3 F, 6 F |
| Case 10 | South | F, 40, CHO | F, 37, Physician | – | – | 10 F |
| Case 11 | South | F, 39, Nurse | M, 40, Physician | – | – | 10 F |
| Case 12 | South | F, 38, CHO | F, 45, Nurse | – | – | 8 F |
| Case 13 | South | F, 41, CHO | F, 48, Nurse | – | – | 6 F |

| Domain | Cadres of healthcare workers | Total Median Interquartile range (IQR) | Part-time Median (IQR) | Number of sites with none |
|---|---|---|---|---|
| Personnel | Doctors | 1 (0-2) | 0 (0) | 6 |
| | Nurse/Nurse-midwife | 4 (1-5) | 1 (0-2) | 3 |
| | Community Health Officer | 1 (0-2) | 0 (0) | 5 |
| | Community Health Extension Workers | 5 (3-7) | 1 (0-3) | 0 |
| | Junior Community Health Extension Workers | 2 (1-3) | 1 (0-3) | 3 |

**Staff NCD service delivery, training and support**

| Domain | Services provided | Northern region PHCs | Southern region PHCs |
|---|---|---|---|
| Hyper-tension service delivery | Screen | 7 | 6 |
| | Diagnosis | 7 | 6 |
| | Prescribe & dispense initial treatment | 6 | 6 |
| | Prescribe follow up treatment | 2 | 5 |
| | Provide long time care | 2 | 5 |

| Domain | Services provided | Northern region PHCs | Southern region PHCs |
|---|---|---|---|
| Diabetes service delivery | Screen | 7 | 6 |
| | Diagnosis | 6 | 6 |
| | Prescribe & dispense initial treatment | 1 | 6 |
| | Prescribe follow up treatment | 1 | 5 |
| | Provide long time care | 1 | 5 |

**Medicines, diagnostics, equipment**

| Domain | | Northern region PHCs | Southern region PHCs |
|---|---|---|---|
| Availability of anti-hypertensive and anti-diabetic drugs | Anti-hypertensive drugs | 6 | 6 |
| | Anti-diabetic drugs | 3 | 6 |
| | Both drugs | 3 | 6 |
| | Neither of the 2 drug categories | 1 | 0 |

| Domain | | Northern region PHCs | Southern region PHCs |
|---|---|---|---|
| Availability of valid NCD rapid diagnostic kit | Valid dipstick for urine protein & glucose | 7 | 6 |
| | Valid dipsticks for urine ketone bodies | 6 | 6 |

| Staff training support | Findings |
|---|---|
| At least, 1 staff trained in management of hypertension and/ or diabetes within the last 2 years | 4 PHCs |
| Availability of treatment guideline or algorithm for hypertension and diabetes | Not available in any PHC |
| Availability of Information, Education and Communication (IEC) on NCDs | Not available in any PHC |

| Domain | Findings |
|---|---|
| Availability of basic NCD Equipment* | All **13 PHCs** have at least one functioning set of basic NCD equipment |
| | **ECG machine** Not available in any PHC |

*(stethoscope, BP apparatus, adult weighing scale, measuring tape, glucometer)

**Fig 1. Summary of NCD-related SARA findings from all 13 PHCs facilities.**

previous 2 years (Table 2). The composition of skilled health workers in PHC facilities varied across states. CHWs constituted most health workers in all the PHC facilities, and were the only health workers in Cases 5, 6 and 7, all of which were northern states. Patients could access care at PHC facilities at any time with or without appointment (Fig 1). PHC facilities, including those with only CHWs provided various services for diabetes and hypertension, ranging from screening, diagnosis, and counselling to drug prescription, referral, and follow-up. However, as a participant in Case 3 said: *"health talk is the number one primary thing that we do at the primary health level, we give health talk to patients and we screen them" (Case 3, FGD, CHEW).*

Knowledge of patient management for hypertension and diabetes varied across PHC facilities and appeared to be related to team capability at each PHC facility. In cases 4, 5, 6 and 7 where the teams comprised mainly CHWs, they were able to obtain clinical history from patients, refer them for laboratory investigations and to a higher-level facility if required:

*"when a patient comes in, we usually take a history . . . . . . (they may) tell you they urinate more than three or four times at night. . . . . . So. . .we usually refer them to the lab for diabetes tests. We diagnose them with diabetes when their blood sugar level is high . . . . . . and refer them"*

*(Case 4, KII, CHO)*

**Table 2.  Disaggregated findings from SARA for NCD services at PHC facilities.**

| Facility | Regional location | Estimated Coverage population (all patients) | % of estimated coverage seen in the last month (all services) | No of in-patient beds (excluding delivery beds) | Availability of at least, one full time physician and nurses | Facility has IEC material on NCDs | Has any NCD guideline or treatment algorithm | ≥1 staff trained (in-service) on NCDs management (in the last 2 years) | ≥1 Basic NCDs equipment | ≥1 Basic NCDs test kits | Stock basic Drugs for HTN/DM | Screen & Diagnose for HTN/DM | Prescribe & dispense initial drugs for HTN/DM | Prescribe and dispense follow-up drugs for HTN/DM |
|---|---|---|---|---|---|---|---|---|---|---|---|---|---|---|
| PHC1 | North | 25,375 | 1.2% | 8 | Nurse | No | No | No | Yes | Yes | Yes | Yes | Yes | Yes |
| PHC2 | North | 10,233 | 1.6% | 0 | Nurse | No | No | No | Yes | Yes | No | Yes | No | No |
| PHC3 | North | 60,375 | 5.7% | 8 | Nurse | No | No | No | Yes | Yes | Yes | Yes | Yes | Yes |
| PHC4 | North | 8,085 | 13.9 | 8 | Nurse | No | No | No | Yes | Yes | Yes | Yes | Yes | Yes |
| PHC5 | North | 12,118 | 8.8% | 14 | Nil | No | No | No | Yes | Yes | Yes | Yes | Yes | Yes |
| PHC6 | North | 9,033 | 4.5% | 16 | Nil | No | No | No | Yes | Yes | Yes | Yes | Yes | No |
| PHC7 | North | 10,382 | 3.7% | 10 | Nil | No | No | No | Yes | Yes | Yes | Yes | Yes | No |
| PHC8 | South | 62,022 | 4.4% | 0 | Physician, Nurse | No | No | Yes | Yes | Yes | Yes | Yes | Yes | Yes |
| PHC9 | South | 44,000 | 3.5% | 0 | Physician, Nurse | No | No | Yes | Yes | Yes | Yes | Yes | Yes | Yes |
| PHC10 | South | 47,808 | 4.0% | 0 | Physician, Nurse | No | No | Yes | Yes | Yes | Yes | Yes | Yes | Yes |
| PHC11 | South | 47,650 | 6.2% | 12 | Physician, Nurse | No | No | Yes | Yes | Yes | Yes | Yes | Yes | Yes |
| PHC12 | South | 3,560 | 9.3% | 8 | Physician, Nurse | No | No | No | Yes | Yes | Yes | Yes | Yes | Yes |
| PHC13 | South | 28,925 | 2.3% | 8 | Nurse | No | No | Yes | Yes | Yes | Yes | Yes | Yes | No |

IEC–Information, Education and Communication; HTN–Hypertension; DM–Diabetes mellitus.

In Cases 8, 9 and 10; all facilities with full-time physicians, prescription, treatment, and follow-up appointments were provided to those diagnosed with hypertension and diabetes–and those with complications were referred to secondary health facilities for further management.

*"For hypertension and diabetes, we just prescribe medications for them and if we have anyone with complications, we refer to general hospital. But the ones we can manage, we give them their routine drugs and see them every two weeks for BP checks and drug refills . . .for diabetic patients yes, when they come, we check their fasting blood sugar every two weeks. . .then we refill their drugs for them*

*(Case 8, KII, physician).*

Insufficient staffing was identified as a major issue. To make up for staffing shortfalls, PHC facilities engaged CHWs as volunteers or contract staff who did not earn salary from the government or receive government support for further training or career progression. This situation was particularly prevalent in the facilities in the northern states and was identified as contributing to lack of motivation: *"the volunteers work to support the hospital, they are trying [their best], but sometimes they will feel "I'm just doing this, after all, at the end of the day what will I get?", you understand. . .it makes one weak [discouraged]"* [Case 3, KII, 1]. The issue of insufficient staff was raised at most PHC facilities, as it led to increased workload and poor service delivery according to a participant in Case 5:

*"It has not been easy, because of lack of staff. Because of the workload, you get [to make] so many mistakes if you are stressed up. . .. . . . at times you find out that you are the only person on duty. . .. and the work is so much you cannot leave . . .. . . that means your family will not see you.*

*[Case 5, KII participant, CHO].*

In Case 6, insufficient staffing also negatively impacted service delivery as CHWs did not have the time to effectively raise awareness for disease prevention and health promotion activities, including that of NCDs:

*"It is due to lack of manpower that we are not able to do all this in the community. The facility is there, patients are there, only one person cannot be [doing everything] . . .. because we have limited CHEWS and JCHEWS, so we need to be in the facility, that is why we are not doing the community activities.*

*[Case 6, FGD].*

By contrast, in the southern states—cases 8, 9 and 10 in particular—physicians operated three shifts daily to provide 24-hour services to patients. Physician attrition and migration abroad was, however, a challenge, as a Medical Officer of Health described in Case 8:

*"in January. . . the government . . .employed doctors . . .but some spent one, two, three months, (and then) they turned in their letter of resignation. . .we have more than fifty percent of doctors that are writing IELTS, PLAB, . . ..to relocate to Canada, UK, US, Caribbean Islands to work, . . . remuneration is part of it".*

*[Case 8, KII1, Medical Officer of Health].*

Insufficient staffing was linked to insufficient training to deliver NCD services. Apart from Cases 8, 9, 10 11 and 13 (all in southern states), no PHC staff had received in-service training on the management of NCDs within the last two years. Most CHWs relied on the knowledge acquired during their training in the College of Health Technology, which they considered inadequate for NCD management: *". . .for hypertension and diabetes. . .the knowledge we are using is the one we got in school [and]. . .we are not updated. . .. There are so many (new) drugs that are in use now, we are using the old knowledge." [Case 5, FGD].* Those who had received some in-service training seemed to have been provided only with skills in NCD screening. In Case 1, a CHW said: *"there was a training we went to, but basically [it] was not for management, but for screening" [Case 1, KII 2, CHW].* Another constraint was that permanent CHWs were usually prioritized and volunteer or contract CHWs were not provided with training opportunities. In Case 6, a CHO said: *"if it is in-service, the government will support you with your salary. . .. but those that are contract staff have no one to support them, that is one of the problems that is preventing (them) from going further" (Case 7, KII, CHO).*

While lack of in-service training for CHWs in NCD management appeared to be a major issue, the presence of a physician in Cases 8, 9, 10, 11 and 12 (southern states) possibly mitigated this issue. This was evident with the use of a more systematic approach in the management of patients with hypertension and diabetes compared to facilities that lack physicians. For instance, in Case 10, classification, stages and systemic features that are pointers to complication for patients with NCDs were mentioned as criteria that guide service delivery:

*". . .if the BP still elevated, then we commence our treatment. . .. . then, we give a close appointment to the patient too. . . . then if it's extremely elevated like a severe hypertension, or hypertensive urgency, we give the patient the first aid, . . .. if the patient comes with other systemic manifestations. . .we administer the first aid, then we refer the patient to a secondary facility"*

*(Case 10, KII participant, Physician).*

## NCD facility supplies and treatment guidelines

All facilities screen for hypertension and diabetics but only 7 provide follow-up and long-term treatment. Blood pressure lowering medication are available in 12 facilities while blood-sugar lowering medications are available in 9 facilities (Table 2). None of the 13 PHC facilities had NCD guideline or information, communication, and education materials on NCD (Table 2). While all the PHC facilities claimed to have basic equipment for screening, diagnosing, and monitoring diabetes and hypertension, their availability did not guarantee functionality. In Case 3, a nurse said *". . . when you enter my office there, there are 2 BP apparatuses. . . you use it [for] a day or two, it will then develop problems. And that is how it continues. (Case 3, KII 1, Nurse-in-Charge).* In Case 6, a CHO said *"We don't have much equipment. . . If you don't have equipment on ground, you will not be able to do your work." (Case 6, KII 1, CHO).*

In all but one facility (Case 2) BP-lowering and blood-sugar lowering medicines were stocked, prescribed, and dispensed. However, the facilities had varying supply and delivery chain structures which helped to ensure that supplies were in place. For example, in a PHC facility included in the Performance Based Financing [PBF] initiative, a CHO said *". . .this facility is a PBF facility, . . . so, they have pharmaceuticals companies that are registered, and you can you buy from any of [those] registered pharmacies." (Case 5, KII1, CHO).* In another example, the supply chain for Cases 8, 9, 10 and 11 used an established state government process with built-in accountability mechanisms. The supply structure, subsidised the costs of medicines and tracked the revenue generated:

"...the State policy regarding essential drugs in all the more than 300 PHC is [such that]... the drugs are procured at the State Medical Store and not over the counter ...all vendors are duly registered at the State level .... and everybody will come together and purchase or procure in the same purse.

(case 8, KII, Medical Officer of Health).

Other PHC facilities adopted an informal practice in their supply system. For instance, in Case 3, a senior nurse was the officer-in-charge of the facility's administrative activities, while a pharmacist was in charge of medicine supply. The arrangement meant that transactions were transparently done through the government's account, with restricted access to potential unauthorized use of internally generated revenue by any facility official. However, the matron-in-charge operated a parallel unofficial supply system with individual pharmaceutical vendors. So, rather than prescribing medicines to be dispensed through the PHC facility's pharmacy, she prescribed and dispensed medicines directly to patients during consultation from her private unofficial supply in exchange of money paid to her but without a receipt.

The supply chain influenced facility operations and team dynamics. For instance, in Cases 1, 4, 5, 6 and 7, a CHW was the officer-in charge–i.e. oversaw the PHC facility's administration which included medication and equipment supply. The CHW generated internal revenue to run the facility by hiring other CHWs on contract and sharing local profits. In some PHCs facilities that have nurses as staff members but a CHW as officers-in-charge of the facility, such as Cases 1 and 4, nurses were mainly in charge of maternity services while the CHWs attended to patients with NCDs and minor ailments. The nurses were protective of their "maternity territory" for two reasons. First, they did not consider the CHWs to be sufficiently competent:

"Well, it was in the north I first found out about CHWs, because in my previous state, we hardly use them, but here they said they are the ones in charge of the community. I will encourage government to employ more doctors to manage the PHC, and let them stick to their primary job, which is immunization, because many of them are being used as doctors in the PHC, which we are not happy about.

(Case 1, KII, Nurse)

Second, because maternity services are revenue-generating services, those who provided the services had greater access to the revenue generated from ante-natal care and delivery services. Hence, this was another reason for role protection, as was highlighted in Case 6, where a CHW said: "we don't have problem with the doctors, but nurses, at times they see this facility as community health workers are taking all the patients and leaving them without any, that's the only problem (Case 6, FGD participant—CHW). The role conflict between nurses and CHWs was enabled by not implementing the defined scope of practice within the PHC facilities.

The Standing Order is the legal document that defines the scope of CHWs' practice. Despite the restriction it places on the management of NCDs by CHWs, many prescribe and dispense NCD medication, monitor NCD patients and refer those they considered complicated condition to secondary health facilities which is beyond their scope of practice. These services are provided without any management guideline or dedicated in-service training. CHWs in all the northern state facilities stated that there were no NCD guidelines or treatment protocols available: "...there is no guideline, the only guide that I will talk of is our standing order... and [there] is no place where our standing order says you should treat hypertension, most of this our standing order, they will say refer" Case 1, FGD, CHW). The non-availability of treatment guideline was also raised as a concern among physician as expressed by one of the participants:

*"No, [no guideline] . . . we should know the new management guideline for managing high blood pressure. . . . It is good for every doctor to be on the same page. . . .if we have a guideline, we'll be on the same page." (Case 8, KII, physician).*

## PHC referral linkages and feedback relations

All PHC facilities had linkage with at least one secondary health facility for patient referral. Participants believed that this helped to ensure continuity of care. In Case 5, a CHW said: *"If the person has diabetes, we refer to general hospital, because we work based on the standing order."* (Case 5, KII 1). And in Case 3, a nurse said: *"normally, we send them to the general hospital, so if they go and they're attended to, they come back here to be checking their BP" (Case 3, FGD).* While PHC facilities in southern states had access to transportation services for referrals to secondary health facilities, the northern state PHC facilities (Cases 1–6) did not: *". . .we don't have any means of referral so if a patient is having a relation, they will go and look for a vehicle [to]. . . come and take the person to the facility" (Case 3, KII).* In one southern state PHC facility (Case 9) patient transportation was available with a functional ambulance for transporting referred patients diagnosed with a serious illness, and they are usually accompanied by a healthcare worker *"So, any patient that we refer, . . .We call our ambulance driver to take the patient to general hospital. . . Yes, one of our staff will follow them to the general hospital" (Case 9, FGD).*

Despite the use of a two-way referral form, feedback on referred patients from secondary health facilities was generally limited. This was partly driven by competing workload at the referral centres: *". . .[for] referral, we [should] get feedback but it's not so. . . The [doctor], maybe his table is full, he cannot even attend to you. I went there third time this week. I had to stay till today to get it" (Case 4, KII1, CHEW).* However, in Case 11, the challenge of feedback appeared to be related to secondary health facilities 'looking down' on the PHC facilities as less important and not deserving of feedback. *". . .they look down on PHC workers as quacks, and so, they feel reluctant [to write feedback] . . .they say, "there's no need of writing back to them, they don't know what they're doing" (Case 11, KII1, Physician).* In Case 6, participants (CHWs) said, not only do physicians and nurses in secondary facilities see PHC as a second rate, but they also deride their services in front of patients and their relatives:

> *"When we refer, they normally condemn our services. These are things they are not supposed to disclose to the patients. . . Even if we do it wrong, they could have called us and tell us, but they will go directly to the patient and tell the patient that we do not know what we are doing, why did you even go there*
>
> *(Case 6, FGD).*

Despite the challenges, staff from some PHC facilities (Cases 4, 5, 6 and 7) managed to obtain feedback for referred patients by visiting the secondary health facilities. The motivation to do so was driven by the Performance-Based Financing (PBF) program that gives financial incentives to PHC facilities for every referred patient with documentary evidence of feedback. According to a participant in Case 4, referral would take place even if there was no physician available: *". . .that [the availability of a physician] will not stop us from referring [to secondary facilities] . . .. because the referral service is in our [PBF] checklist, and it is also a source of money for us." (Case 4, KII participants, CHEW).* The utility of the obtained feedback in the continuum of care for the patients was difficult to ascertain. However, obtaining feedback was easier in Cases 8, 9 and10 where the patient was referred between physicians. For instance, one site (Case 10) had an established social media platform for communication between physicians at PHC and secondary health facilities without connotations of inferiority:

*". . .there's a new development going on now, we actually had a meeting, because we were having issues with referring patients to them . . .. so, they came up with an idea of creating a WhatsApp group, whenever we refer a patient, we call them, we call them before referring. . . then we will also post on the group about the patient we referred, so, they will give us feedback [on the platform]"*

*(Case 10, KII1, Physician)*

Administrative feedback from the Local Government Area office occurred in response to data submitted from PHC facilities. Patient information were mainly held on paper records and entered into a daily register. Selected NCD cases, along with some other infectious diseases, were entered into the Monthly Health Facility Summary Form and Integrated Diseases Surveillance for onward transmission to higher level of the health system. According to a participant in Case 5, feedback given to the PHC facility mainly focused on data quality with little attention on quality of services public health actions that need to be taken:

*"For hypertension we send it [data] to the Local Government Area (LGA) office at the end of the month, the LGA now send it to the state, the state now sends it to national. . .. they will count again from the summary and the register to see whether there are discrepancies, so and if there are, they will now correct it. . . They will write a report and give you, areas where you need to improve and areas that you have done well.*

*(Case 5, KII1, CHO)"*

There were also community referral linkages. The PHC system operates within the Ward Health System structure, which provides direct linkage to the community, helps to focus the PHC facility on their needs, raise awareness about government programmes, and mobilise community members to participate. Community outreaches (part of routine PHC services) were therefore an avenue to diagnose undetected diabetes and hypertension:

*"So, you see people coming from the community. That's where we pick the highest number of high blood pressure cases, . . .we give them drugs for free and maybe ask them to come back for follow-up. So, subsequently they come for follow-up, and we write drugs for them, but they have to pay.*

*(Case 8, KII3, Physician)"*

These community outreaches sometimes received ad-hoc support from non-governmental and commercial organizations which sometimes partner with PHC facilities to conduct free screening for hypertension and diabetes in the community and subsequently linked newly diagnosed patients to the PHC facilities for further management:

*"Once in a while you'll see some NGO and a particular commercial bank will come, [and say], 'we want to do hypertension and diabetes'; we allow them; we call people; we move around, call people. . . They brought some drugs and the BP apparatus, and we joined them. They also came with their health workers (including doctors)*

*(Case 8, KII2, Nurse).*

Fig 2 shows patient flow for NCD service delivery at the PHC level, with enablers and barriers along the pathway which we describe below according to the identified themes.

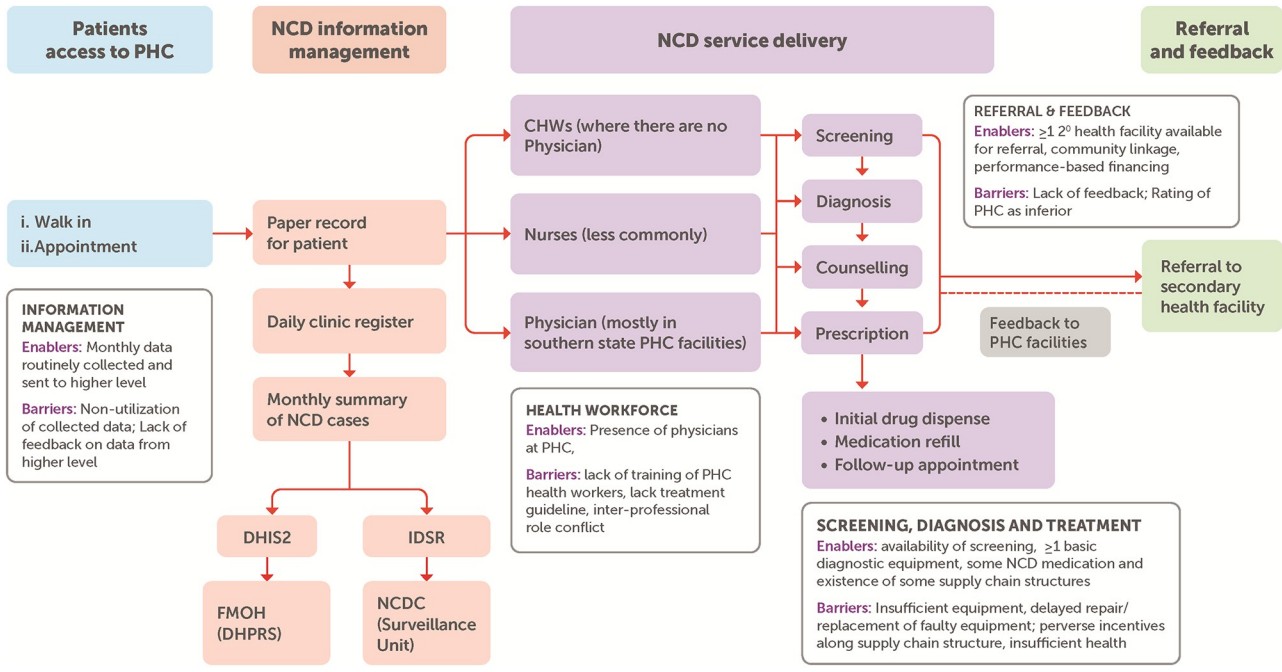

**Fig 2. Patient flow for NCD service delivery at the PHC level, with enablers and barriers along the pathway.**

## Discussion

This mixed methods case study of 13 PHC facilities in Nigeria examined the organisation of PHC service delivery for NCDs in Nigeria. It provides insight into a range of factors that serves as potential enablers and barriers to delivery of care for NCDs. Key findings include: (1) role conflict among non-physician health workers; (2) Inadequate PHC workforce, and perception of PHC as inferior; (3) the role of the physician as catalyst for NCD service delivery (4) the use of perverse incentives to sustain the functioning of PHC facilities, and (5) the variation in PHC service delivery by geographical region. We discuss each of these in detail below.

### Interprofessional role conflicts

The delivery of quality health care is dependent on the contributions from the various cadres of healthcare workers that constitute the team. This involves a complex process, particularly at the PHC level where care needs of patients and service delivery environments are diverse [30]. Interprofessional role conflicts among the healthcare team member arise from this complexity with potential to decrease quality care for patients and reduce team effectiveness [31]. Possible sources of such conflicts could be substantive (particularly in relation to scope of practice and financial renumeration) and emotive (particular when driven by individual personalities and entrenched power differentials) [32]. In our study, non-availability of guidelines, lack of clear roles, control of internally generated incomes and the display of professional cadre superiority were some issues identified as potential sources of role conflicts among the non-physician PHC team members.

Previous studies, from high income countries, have discussed interprofessional role conflicts among PHC teams as a barrier to quality care–with discrepancies in guidelines [33], scope of practice, role boundary issues and accountability among the issues identified as sources of conflicts [30, 34]. Similar study from a LMIC setting revealed interprofessional role

conflict among non-physician health workers providing care for patients with cardiovascular diseases [35]. PHC facilities in Nigeria are characteristically staffed with CHWs and nurses [36], with the latter often in the majority [14], it is important that conflict resolution strategies that support individual and the team as a whole are put in place. These include the development of an explicit, contextualized guideline with clear role description, standardised facility revenue management processes, task-shifting and task-sharing programmes [37, 38], and effective leadership [39].

## Inadequate PHC workforce, and the perception of PHC as an inferior component of the health system

An adequate health workforce in terms of supply and skill mix is essential for effective NCD service delivery. A major barrier observed in this study is that most PHCs facilities do not meet the minimum staffing requirements stipulated by the Ward Health System within which each PHC operates [10]. This persistent workforce shortage combined with a large pool of volunteer community health workers likely reflects limited funding allocations for the PHC sector. National and subnational governments need to accelerate and scale up staff recruitment combined with appropriate skills training [40], and proportionate distribution to areas of relative low workforce density [41]. This requires development of evidence-informed workforce policies, effective deployment mechanisms [41] and increased funding allocation.

Being the closest entry point of the community into the formal health system, the PHC sector is best placed to reach the 'last mile population' who predominantly reside in rural areas [12]. The existence of established referral linkages between the PHC and secondary facilities is considered an enabler for the management of patient with NCDs. Referrals are aimed at ensuring patients receive the appropriate quality and continuity of care within the health system [42]. However, referral can only be effective if there exists a close and congruent relationship among the various levels of care within the health system [43]. The labelling and perception of the PHC level, and its staff, as inferior rather than as a partner-in-care within the health system by some secondary health facilities is a major challenge. A reason for this attitude identified in our study was the cadre of health care workers that make up the PHC staff. We found this led to delayed or absolute lack of feedback from secondary health facilities and this has been observed in other studies [44, 45]. The impact of a one-way referral system without feedback to the PHC is disruption to the continuum of care for a patient [46].

Previous studies also identified weak referral linkage for NCD management between primary and secondary health facilities as a barrier to high quality service delivery [47]. It is important that governments at every level address this issue. Enacting and implementing appropriate referral policies, training of health workers, and activities to promote interprofessional collaboration accordingly could address health worker attitudes and reorientate secondary health care workers on the importance of feedback [46]. Appropriate use of referral guidelines could also help to clarify areas of disagreement between different levels of health system, halt or reverse personnel's view of PHC as second-rate care and respectfully acknowledge PHC workers as members of the healthcare team [48].

Conversely, a promising enabler of referrals and linkages to the community identified in our study was the interest of civil society organizations (CSOs) in supporting NCD services, at the PHC level–and especially in supporting community-based outreaches that promote diagnosis and referrals. This finding is in line with previous studies in Nigeria that show the role and potential of community efforts (via community health committees) in supporting the day-to-day functioning of PHC services, linking community members to PHC facilities, and building community trust in PHC services [49, 50]. Such examples of social collaboration promote

community participation, with potential to empower PHC workers in prioritizing the needs of community members within this space of PHC-CSO/community co-responsibility [51].

## Physician at PHC facilities as catalyst for NCD service delivery

Aside the private sector, physicians are not typically seen at PHC settings in sub-Saharan Africa [52]. Where they are found, they tend to enable high quality of care and support management of complex NCD cases, providing continuous quality care with resultant cost reductions for both the patient and health system as well as increasing community trust in the PHC system [53]. The presence of physicians in PHC facilities, as observed in this study, can therefore be considered an enabler to quality NCD service delivery. In addition to more advanced care for patients, PHC physician referral and feedback may also be enhanced among colleagues at the secondary health facilities. However, it is important that physician-centric biomedical care delivery models are avoided. This requires physician's roles to be expanded beyond direct clinical care for patients [54]. Such roles may include managerial and administrative roles, formal and on-the-job training roles, and supportive supervision for the health workforce team.

Despite their important role, employment of physicians at PHC facilities is not realistic in most states of Nigeria due to critical shortage, maldistribution (between rural and urban facilities, and between northern and southern states) and migration (out of the country) [36]. It is therefore important that, in PHC facilities with physicians, effective strategies and structures be put in place to limit the risk of physician attrition. Where employment of physicians is not feasible, there is the need for effective implementation of task-shifting and task-sharing with nurses and CHWs. The current situation in which tasks are shifted to nurses and CHWs will need to be transformed to one in which task-shifting and task-sharing is deliberate and supported with widely disseminated and regularly updated decision-support job aids. This has potential to improve NCD management as was done successfully with maternal health, HIV [55] and contraception service provision [37, 38].

## "Perverse incentives" for sustaining the functioning of PHC facilities

There is a general perception that health system funding is inadequate, and PHC facilities are the worst hit by this [56]. Multiple government agencies are involved in the financing of PHC service delivery but local governments are primarily responsible for the funding the day-to-day functioning of the health facility [57, 58]. Due to inconsistent, insufficient, or absolute non-release of funds, most local governments are unable to support PHC facilities beyond payment of salaries [14]. This may be one of the reasons why the management of various PHC facilities have devised unofficial or informal means as a workaround strategy to generate revenue internally for sustaining the daily function of the PHC facilities.

Such informal practices have previously been documented as often essential for the day-to-day functioning of PHC facilities in Nigeria [49, 50]. As shown in our study, for instance, funds generated from unauthorized sales of drugs have been used to engage and retain unemployed health workers on a contract basis, so as to make up for staff shortfalls. While this can superficially be judged as corruption, those directly affected may argue that it is a rational adaptation to the existing PHC governance environment where funding is heavily constrained to meet community need. If funding and human resources at these facilities remain the same, efforts to control or police perverse incentives (which help to sustain the functioning of otherwise sub-optimally supported PHC facilities) are likely to prove impractical.

## Intra-regional variation

A key finding from our study was the distinct delineation of functions and structures between PHC facilities in northern as compared southern states in Nigeria. This was evident in terms of staffing capacity, cadres of health workers in the PHC teams, training, medication availability and supply chain structures. This disparity of PHC facilities' functionality across geographical zones, is also reflected in wide disparities in service provision–for example, 0.5% of PHC facilities in the northeast Nigeria provides immunisation services compared to 90% in southwest Nigeria [14]. Much of the inequalities have been linked to historical, political, ethno-religious and socioeconomic reasons [25]. It is estimated that over 70% of Northern population lives below poverty line compared to less than 35% in the South [59].

The uneven distribution of doctors, about 160 per million population in the North and 443 per million populations in the South have also been attributed to financial, conflict and social reasons [60]. Another possible explanation for these disparities is the greater capacity for public financing of health in the southern states as a result of disproportionate economic development when compared to the northern states [61]. This disparity reflects the situation of the health workforce in northern states where CHWs are more likely to be in charge of PHC facilities than nurses or physicians. Although we observed relative differences between north and south states, the absolute lack of physician-run PHC facilities is a national problem, as is the use of a large pool of volunteer CHW staff, a phenomenon that threatens service delivery quality, continuity, and sustainability not only for NCDs but for other PHC services.

An important policy direction for PHC in Nigeria is the development of new state-level PHC agencies to take over PHC governance from both local and national governments [62]. On the one hand, the state-level PHC agencies will centralise PHC management within a state by taking over responsibilities from local government councils–given their weak financial and technical capacity in most states. On the other hand, state-level PHC agencies may also take over direct policy guidance from the national PHC agency–thereby better tailoring PHC policies to local needs and circumstances [63].

Our study shows different models of PHC (CHEW vs Physician led) which vary significantly by context: northern states vs southern states; and rural communities vs urban communities. Our findings can inform the ongoing efforts of state PHC agencies to re-organise NCD service delivery in a way that not only reflects their greater capacity relative to local governments, but also reflects greater contextualisation to local needs and capacity.

## Limitations

While the findings of this study are not generalisable to all PHC facilities in Nigeria, the facilities included represent some of the geographical variation across the country. The findings provide a textured and contextualized understanding of the organisation of PHC services for NCDs in Nigeria drawing on a broad range of perspectives and observations. More so, our study focused primarily on the operational structure within which services are being delivered for NCDs. However, a limitation is that this study did not address patient care directly which would have been desirable to understand the perception of care recipients, and what may constitute barriers and enables of NCD services at the PHC level from their perspective. We recommend that future studies should explore PHC service organisation with a focus on the socioeconomics, political and governance structures that apply to NCD service delivery.

## Conclusion

Our study highlighted essential considerations in efforts to strengthen the PHC system for NCD service delivery in Nigeria. Priority considerations include: (1) Adequate funding and

staffing of the PHC system to ensure optimal health workforce strength considering the regional, socio-political and economic variations. (2) Continuous capacity building of PHC health workers with focus on NCD prevention and management. (3) Implementing task-sharing and task-shifting policies for NCDs among non-physician health workers, with clear role delineation and promotion of inter-professional networks and collaboration (4) Development of NCD treatment guidelines and protocols, and making them available and accessible at PHC facilities, adapted to the cadre and mix of the workforce available at each PHC facility (5) Financial and technical investment into basic NCD equipment, essential NCD medicines (with the essential medicine list at the PHC level revised to reflect this) and medicine supply chain structures. (6) Strengthening referral linkages between PHC and higher-level facility, and between communities and PHC facilities while also effectively integrating NCDs services into existing PHC structures. We acknowledge that all these need to take place within the appropriate political and technical leadership that govern the PHC system.

## Supporting information

**S1 Annex. Health facilities summaries from quantitative and qualitative data.**
(DOCX)

**S2 Annex. MS Excel spreadsheet summary on SARA findings.**
(XLSX)

**S1 Questionnaire. Questionnaire on 'inclusivity in global research'.**
(DOCX)

## Acknowledgments

The authors acknowledge the support of the Australian Commonwealth Government provided to WSA through the Australian Government Research Training Program Scholarship. We express our sincere gratitude to all health agencies, study participants and research assistants who supported us through the data collection period.

## Author Contributions

**Conceptualization:** Whenayon Simeon Ajisegiri, David Peiris, Rohina Joshi.

**Data curation:** Whenayon Simeon Ajisegiri.

**Formal analysis:** Whenayon Simeon Ajisegiri, Seye Abimbola, Azeb Gebresilassie Tesema, Olumuyiwa O. Odusanya, David Peiris, Rohina Joshi.

**Funding acquisition:** Rohina Joshi.

**Investigation:** Whenayon Simeon Ajisegiri.

**Methodology:** Whenayon Simeon Ajisegiri, Seye Abimbola, David Peiris, Rohina Joshi.

**Project administration:** Whenayon Simeon Ajisegiri, David Peiris, Rohina Joshi.

**Resources:** Whenayon Simeon Ajisegiri, Seye Abimbola, David Peiris, Rohina Joshi.

**Supervision:** Seye Abimbola, Olumuyiwa O. Odusanya, David Peiris, Rohina Joshi.

**Visualization:** Whenayon Simeon Ajisegiri, Seye Abimbola, Azeb Gebresilassie Tesema, Olumuyiwa O. Odusanya, David Peiris.

**Writing – original draft:** Whenayon Simeon Ajisegiri.

**Writing – review & editing:** Whenayon Simeon Ajisegiri, Seye Abimbola, Azeb Gebresilassie Tesema, Olumuyiwa O. Odusanya, David Peiris, Rohina Joshi.

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
