## [Decision Letter · Decision Letter 0]

21 Apr 2022

PGPH-D-22-00322

The organisation of primary health care service delivery for non-communicable diseases in Nigeria: A case-study analysis

Dear Dr. Ajisegiri,

Thank you for submitting your manuscript to PLOS Global Public Health. After careful consideration, we feel that it has merit but does not fully meet PLOS Global Public Health’s publication criteria as it currently stands. Therefore, we invite you to submit a revised version of the manuscript that addresses the points raised during the review process.

We look forward to receiving your revised manuscript.

Kind regards,

Roopa Shivashankar, MD, MSc

Academic Editor

Journal Requirements:

1. Please include a complete copy of PLOS’ questionnaire on inclusivity in global research in your revised manuscript. Our policy for research in this area aims to improve transparency in the reporting of research performed outside of researchers’ own country or community. The policy applies to researchers who have travelled to a different country to conduct research, research with Indigenous populations or their lands, and research on cultural artefacts. The questionnaire can also be requested at the journal’s discretion for any other submissions, even if these conditions are not met.  Please find more information on the policy and a link to download a blank copy of the questionnaire here: https://journals.plos.org/plosone/s/best-practices-in-research-reporting. Please upload a completed version of your questionnaire as Supporting Information when you resubmit your manuscript.

2. Your co-authors:

David Peiris -dpeiris@georgeinstitute.org

Rohina Joshi -rohina.joshi@unsw.edu.au

,have not confirmed authorship of the manuscript. We have resent them the authorship confirmation email; however please check that the above email address for them is correct and follow up personally to ensure they confirm. 

Please note that we cannot proceed your manuscript  until we have received confirmations from all co-authors.

3. Please include details in the Funding Information section in the system. It should matched with the Financial Disclosure Statement.

4. We have noticed that you have uploaded supporting information but you have not included a list of legends.  Please add a full list of legends for all supporting information files (including figures, table and data files) after the references list. 

5. In the online submission form, you indicated that "All relevant data contributing to the findings are within the study. The raw data (transcript from qualitative study) are stored on a secure network and can not be made publicly available in order to protect participant confidentiality (Making them publicly available also contradict the terms contained in the ethical approval for the study).". All PLOS journals now require all data underlying the findings described in their manuscript to be freely available to other researchers, either 1. In a public repository, 2. Within the manuscript itself, or 3. Uploaded as supplementary information.

Additional Editor Comments (if provided):

Reviewers' comments:

Reviewer's Responses to Questions

**Comments to the Author**

1. Does this manuscript meet PLOS Global Public Health’s publication criteria? Is the manuscript technically sound, and do the data support the conclusions? The manuscript must describe methodologically and ethically rigorous research with conclusions that are appropriately drawn based on the data presented.

Reviewer #1: Yes

Reviewer #2: Yes

2. Has the statistical analysis been performed appropriately and rigorously?

Reviewer #1: Yes

Reviewer #2: Yes

3. Have the authors made all data underlying the findings in their manuscript fully available (please refer to the Data Availability Statement at the start of the manuscript PDF file)?

Reviewer #1: Yes

Reviewer #2: Yes

4. Is the manuscript presented in an intelligible fashion and written in standard English?

Reviewer #1: Yes

Reviewer #2: Yes

5. Review Comments to the Author

Reviewer #1: The paper highlights some very important issues about the PHC system in Nigeria. It needs minor edits and additions.

Please clarify this quote- 'consequently, NCD care at the PHC level is mainly handled by community health workers (CHWs), who are not generally lack sufficient training in NCD management and prevention [12]'.

The the issue of referral linkages was highlighted but not discussed- PHC should be the entry point into the health system and so this should be addressed.

It is worth discussing that the study has shown that different models of PHC (CHEW vs Physician led) and different contexts (North vs South; Rural vs Urban) were assessed therefore further contextualisation of the findings will make the recommendations more useful by policy makers.

Some background on the policy direction of the PHC system in Nigeria wil also provide some useful background for the readers of the paper.

Reviewer #2: This is a well written paper highlighting the major challenges to manage non-communicable diseases in Nigeria at primary health care level. There are few suggestions which authors may consider.

1. Justification for selecting health workers who have worked for a minimum of three

months at the facility in the study

2. Insufficient staffing is the major issue in result section but it is not discussed in discussion nor it is part of figure 2.

3. Community linkage is another enabler from and Performance-Based Financing (PBF) program are another enabler discussed which can be added in figure 2 under referral heading.

6. PLOS authors have the option to publish the peer review history of their article (what does this mean?). If published, this will include your full peer review and any attached files.

**Do you want your identity to be public for this peer review?** For information about this choice, including consent withdrawal, please see our Privacy Policy.

Reviewer #1: **Yes: **Emmanuel Agogo

Reviewer #2: No

---

## [Decision Letter · Decision Letter 1]

31 May 2022

The organisation of primary health care service delivery for non-communicable diseases in Nigeria: A case-study analysis

PGPH-D-22-00322R1

Dear Dr Ajisegiri,

We are pleased to inform you that your manuscript 'The organisation of primary health care service delivery for non-communicable diseases in Nigeria: A case-study analysis' has been provisionally accepted for publication in PLOS Global Public Health.

Best regards,

Roopa Shivashankar, MD, MSc

Academic Editor

Reviewer Comments (if any, and for reference):

Reviewer's Responses to Questions

**Comments to the Author**

1. If the authors have adequately addressed your comments raised in a previous round of review and you feel that this manuscript is now acceptable for publication, you may indicate that here to bypass the “Comments to the Author” section, enter your conflict of interest statement in the “Confidential to Editor” section, and submit your "Accept" recommendation.

Reviewer #1: All comments have been addressed

2. Does this manuscript meet PLOS Global Public Health’s publication criteria? Is the manuscript technically sound, and do the data support the conclusions? The manuscript must describe methodologically and ethically rigorous research with conclusions that are appropriately drawn based on the data presented.

Reviewer #1: Yes

3. Has the statistical analysis been performed appropriately and rigorously?

Reviewer #1: Yes

4. Have the authors made all data underlying the findings in their manuscript fully available (please refer to the Data Availability Statement at the start of the manuscript PDF file)?

Reviewer #1: Yes

5. Is the manuscript presented in an intelligible fashion and written in standard English?

Reviewer #1: Yes

6. Review Comments to the Author

Reviewer #1: (No Response)

7. PLOS authors have the option to publish the peer review history of their article (what does this mean?). If published, this will include your full peer review and any attached files.

**Do you want your identity to be public for this peer review?** For information about this choice, including consent withdrawal, please see our Privacy Policy.

Reviewer #1: **Yes: **Dr Emmanuel Agogo
